# Biological impact of lead from halide perovskites reveals the risk of introducing a safe threshold

Junming Li[1,2,3,6], Hai-Lei Cao[1,6], Wen-Bin Jiao[1], Qiong Wang[2], Mingdeng Wei[3]*, Irene Cantone [4,5], Jian Lü [1]* & Antonio Abate[2,3]*

Regulations currently in force enable to claim that the lead content in perovskite solar cells is low enough to be safe, or no more dangerous, than other electronics also containing lead. However, the actual environmental impact of lead from perovskite is unknown. Here we show that the lead from perovskite leaking into the ground can enter plants, and consequently the food cycle, ten times more effectively than other lead contaminants already present as the result of the human activities. We further demonstrate that replacing lead with tin represents an environmentally-safer option. Our data suggest that we need to treat the lead from perovskite with exceptional care. In particular, we point out that the safety level for lead content in perovskite-based needs to be lower than other lead-containing electronics. We encourage replacing lead completely with more inert metals to deliver safe perovskite technologies.

[1] Fujian Provincial Key Laboratory of Soil Environmental Health and Regulation, College of Resources and Environment, Fujian Agriculture and Forestry University, Fuzhou, Fujian 350002, P.R. China. [2] Helmholtz-Zentrum Berlin für Materialien und Energie, Kekuléstrasse 5, 12489 Berlin, Germany. [3] Fujian Provincial Key Laboratory of Electrochemical Energy Storage Materials, Fuzhou University, Fuzhou, Fujian 350002, P.R. China. [4] Dipartimento di Medicina Molecolare e Biotecnologie Mediche, University of Naples Federico II, Via Pansini, 5, 80131 Naples, Italy. [5] CNR Istituto di Endocrinologia e Oncologia Sperimentale (IEOS), Via Pansini, 5, 80131 Naples, Italy. [6] These authors contributed equally: Junming Li, Hai-Lei Cao. *email: wei-mingdeng@fzu.edu.cn; jian_lu_fafu@163.com; antonio.abate@helmholtz-berlin.de

Halide perovskites are the golden boy of the next generation of solar cells, light-emitting diodes, and sensors. A lucky combination of unique optical and electronic properties together with cost-effective processing created the basis for a perovskite revolution. Perovskite-based solar cells (PSCs) are on their way to mass commercialization as standalone technology and in tandem with silicon solar cells for both large-scale energy production and portable electronics. Today, the best performing and the more stable PSCs make use of lead salts, which can pollute the environment with a dramatic impact on human health[1–5]. Even the most rigorous encapsulation and the strictest recycling procedures cannot exclude the risk of leaking halide perovskites into the environment during the life cycle of solar cells and other optoelectronics[6,7]. Organic and inorganic lead salts comprising halide perovskites are relatively soluble in water, which makes them potentially bioavailable, i.e., accessible to plants and consequently other living organisms[8–11]. The actual risk associated with potential soil contamination remains so far unknown. The scientific community is raising concerns about the environmental impact of halide perovksites[9,12], but the argument remains under debate because systematic investigations are not available.

Here, we make use of plants to asses for the environmental impact of lead from halide perovskite contaminating the soil. We show that the safety level for lead content in PSCs needs to be lower than other lead-containing electronics. We encourage replacing lead completely with more inert metals to deliver a safe PSC technology.

## Results

**Design strategy**. The lead was almost absent on earth's crust, but it was readily accessible from the underground. Today, the lead concentration on earth surface is non-zero because humans have been using lead in several applications polluting the earth surface with a large variety of lead compounds.[13] The current production of lead worldwide is depicted in Fig. 1a. For this study, we collected natural soil in China, and we measured an effective lead concentration ~36 mg kg$^{-1}$ (Supplementary Table 1), which is in line with the value reported in several agricultural lands world-wild[14]. An archetypal PSC module made of 0.6-µm-thick methylammonium lead iodide perovskite layer would contain ~0.8 g of lead per square meter; if we disperse the entire lead content of the PSC module on the same area of ground below, the concentration of lead into the soil will increase ~4.0 mg kg$^{-1}$ (detailed calculation are shown in Supplementary Note 1). In China, where the more significant production of lead is concentrated, the agricultural regulation tolerates a lead content into the soil up to 250 mg kg$^{-1}$ [15]. We note that this is also one of the most restrictive lead regulation in agriculture (see Supplementary Table 2). At this point, one is tempted to conclude that the amount of lead in a perovskite solar module is too low to create any significant risk for the environment. Furthermore, the existing regulations about the use of heavy metals in electronics support such a conclusion as the content of lead in perovskite modules is estimated lower than 0.1% by weight, which is the safety limit imposed by many countries worldwide. Therefore, it seems we should not worry about the use of lead in halide perovskites for solar cells and similar optoelectronic applications.

We recreated in the laboratory the scenario described above, that is growing plants into the perovskite-contaminated soil. Plants are the main link that transfers heavy metals from ground to food and thus the human body. The ability of plants to uptake heavy metals from the ground, i.e., the bioavailability of metals, is influenced by pH, the metal concentration, metal cation exchange capability between roots and soil, and many other factors[17].

Agricultural standards indicate the use of mint plants (*Mentha spicata*) to test the lead contamination[18,19]. We collected natural soil to prepare three sets of samples with different concentrations of lead halide perovskites. The first set of samples were made by contaminating natural soils with 5.0 mg kg$^{-1}$ of lead to simulate the scenario described above for a solar module leaching the entire content of perovskite into an equivalent area of ground. For the second set of samples, we used 35.0 mg kg$^{-1}$ of lead to double up the natural content of lead in the soil. Finally, we prepared a collection of samples with 250 mg kg$^{-1}$ of lead, as this is the limit imposed for agricultural soil in China[15]. Furthermore, we explore the difference between the contamination from halide perovskites and PbI$_2$, which is only one of the precursors commonly used to prepare halide perovskites as a source of lead.

Besides the mint plants experiment, we performed additional analysis based on *Capsicum annuum* (chilli), which has low lead accumulation ability, and *Brassica campestris* (cabbage), which has middle lead accumulation ability. Both the chilli and cabbage experiments showed a trend similar to that of mint plants (see Supplementary Fig. 8 and Supplementary Note 2).

**Lead bioavailability**. Fig. 2 shows the lead content measured in the mint plants after 20 days of growth, which is the typical harvest period for mints. As expected from literature data, the lead concentration is higher in the roots and leaves than in the stems. For the mints grown in the natural soil, which account for 36.3 mg kg$^{-1}$ of lead contamination, not from perovskite, the mean lead concentration was 12.0, 3.7, and 8.0 mg kg$^{-1}$ in roots, stems and leaves, respectively. By adding 5.0 mg kg$^{-1}$ of lead from perovskite, the mean lead concentration increased to 26.9, 4.7, and 10.0 mg kg$^{-1}$ in roots, stems and leaves, respectively. The importance of this numbers becomes more clear if we consider that by adding 5.0 mg kg$^{-1}$ of lead perovskite we increased the effective lead concentration of natural soil by only ~10%, whereas the level of lead measured into the plants increased over 100%. Adding 35.0 and 250 mg kg$^{-1}$ of perovskite, we observed an even more pronounced difference between the plant's growth in native or perovskite-contaminated land. In particular, after adding 250 mg kg$^{-1}$ (the safety level indicated from agriculture regulation in China) of perovskite in soils, most of the mint plants showed blackening and rotting, which are indicative of lead intoxication. The few plants that survived did not provide sufficient material for statistical analysis. However, one of the few samples collected reported an extremely high concentration of lead with 3905.8, 209.6, and 405.7 mg kg$^{-1}$ for the roots, stems, and leaves, respectively.

We use the following Equation to describe the mint plants uptake ability as a function of the lead perovskite concentration

$$\eta = 100\% * \frac{[\text{Plants(pervskite soil)} - \text{Plants(natural soil)}]}{\text{added perovskite}} \quad (1)$$

where $\eta$ is the lead uptake coefficient (%); Plants (perovskite soil) the mean lead concentration in plants grown in perovskite-contaminated soils; Plants (natural soil) the mean lead concentration in plants grown in native land, and added perovskite the amount of lead from perovskite added into the native soil. We found the uptake ability of lead increases with the concentration of perovskite into the ground, as depicted in Fig. 2d.

Screening the effect of organic or inorganic cations comprising the perovskite on the bioavailability of lead is of interest for evaluating the environmental impact of different perovskite formulations. We repeated the experiment described above contaminating the soil with 5.0 mg kg$^{-1}$ of lead from PbI$_2$, which is only one of the precursors of the perovskite[20], to investigate the influence of the organic cations employed in the most commonly

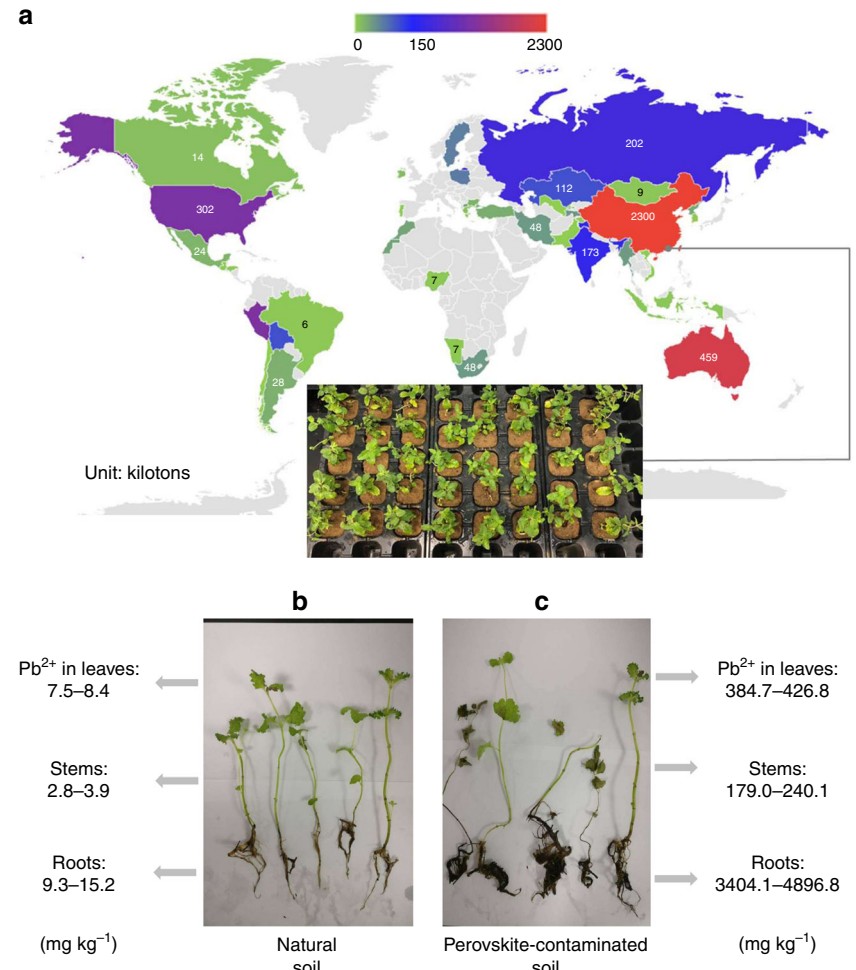

**Fig. 1 Overview of the experiment of plants grown in perovskite-contaminated soils. a** Distribution of lead production worldwide in 2017 (open data from British Geological Survey[16]). The inset shows the photo of mint plants grown in perovskite-contaminated soil within the campus of Fujian Agriculture and Forestry University, China (latitude 26.084, longitude 119.238). **b** The picture of mint plants grown on control soil (left) and **c** 250 mg kg⁻¹ Pb²⁺ perovskite-contaminated soil (right). The range of lead content measured in the leaves, stem, and root is reported on the side of each picture.

used perovskite formulations for PSCs. Not surprisingly, we found that the organic cation has a substantial impact on the lead uptake ability of mint. The data reported in Fig. 2 show that the lead content in mint grown in soil contaminated with $PbI_2$ is similar or even lower than in plants grown in natural soil. We rationalized this result, considering that the organic cation influences the pH of the soil, which has a substantial impact on lead bioavailability.

**Tin bioavailability.** Alternative lead-free perovskite compositions have been proposed to eliminate the risk of lead contamination. Among the others, tin-based perovskites are one of the most exciting possibilities.[21] Although tin can be more toxic than lead for human and animals, upon dispersion in the environment, the $Sn^{2+}$ from perovskite oxidizes rapidly to stable oxygenated $Sn^{4+}$ compounds with low solubility in water.[22] The low water solubility should reduce bioavailability. We repeated the same set of experiments reported for lead-based perovskite to estimate the relative bioavailability of tin compared with lead in halide perovskites. For the mints grown in the natural soil, the mean value of tin concentration in roots and stems were ~4.1 and 2.4 mg kg⁻¹, respectively. In leaves grown in the native soil, the tin concentration was below the detection limit (<1.6 mg kg⁻¹). After adding 5.0 mg kg⁻¹ $Sn^{2+}$, the average tin concentration of roots, stems, and leaves increased to 6.5, 3.6, and 2.0 mg kg⁻¹, respectively. After adding 35.0 mg kg⁻¹ $Sn^{2+}$, the pants uptake more tin than that at 5.0 mg

kg⁻¹ $Sn^{2+}$, which resulted in 17.3, 4.1, and 3.7 mg kg⁻¹ for roots, stems, and leaves, respectively. These values are far below the maximum tolerable levels proposed by the Food and Agriculture Organization of the United Nations[23], which indicates 150 mg kg⁻¹ in canned beverages and 250 mg kg⁻¹ in other canned foods (for certain meat products, the maximum tolerable tin concentration is 50.0 mg kg⁻¹). Moreover, applying Eq. 1, we found that the tin uptake ability decreases with the increased $Sn^{2+}$ soil concentration, see Fig. 3d, which is the opposite we observed with lead in Fig. 2d.

We contaminated natural soil with only $SnI_2$ (not the entire perovskite formulation) to study the influence of the organic cation. After adding 35.0 mg kg⁻¹ $Sn^{2+}$ from $SnI_2$, the mean tin concentration in mints was 12.1, and 3.7 mg kg⁻¹ for the roots and leaves, respectively (blue data in Fig. 3), which are slightly lower than that grown in tin perovskite ($SnI_2$ + MAI) contaminated soil. Therefore, similar to what we found for lead-based perovskite, the organic cations play a role in the bioavailability of the heavy metals. Further studies along these lines are currently under construction.

The mint plants we used in the experiments described above were born with cutting propagation technique and grown with water culture (see Supplementary Note 4), which enabled to investigate the lead and tin uptake ability in mature plants, but not their influence on germination. The latter was investigated using lead and tin perovskite-contaminated soil to grow *B. campestris* from seeds. We observed that both lead and

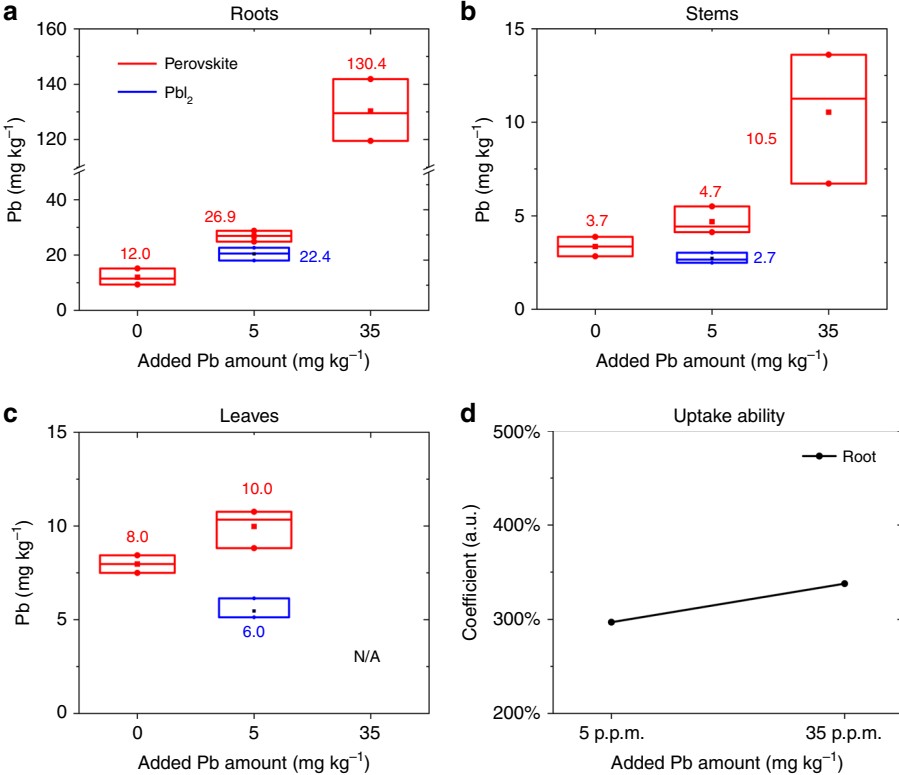

**Fig. 2 Lead concentration in different parts of mint plants. a** Roots, **b** stems, and **c** leaves. The lead uptake ability ($\eta$, defined in Eq. 1) is reported in **d**. Data for lead concentration in leaves at 35.0 mg kg$^{-1}$ are not available (N/A) as plants did not produce enough leaves at such a high level of lead. The maximum, minimum, and mean value of each distribution are reported in **a**–**c**.

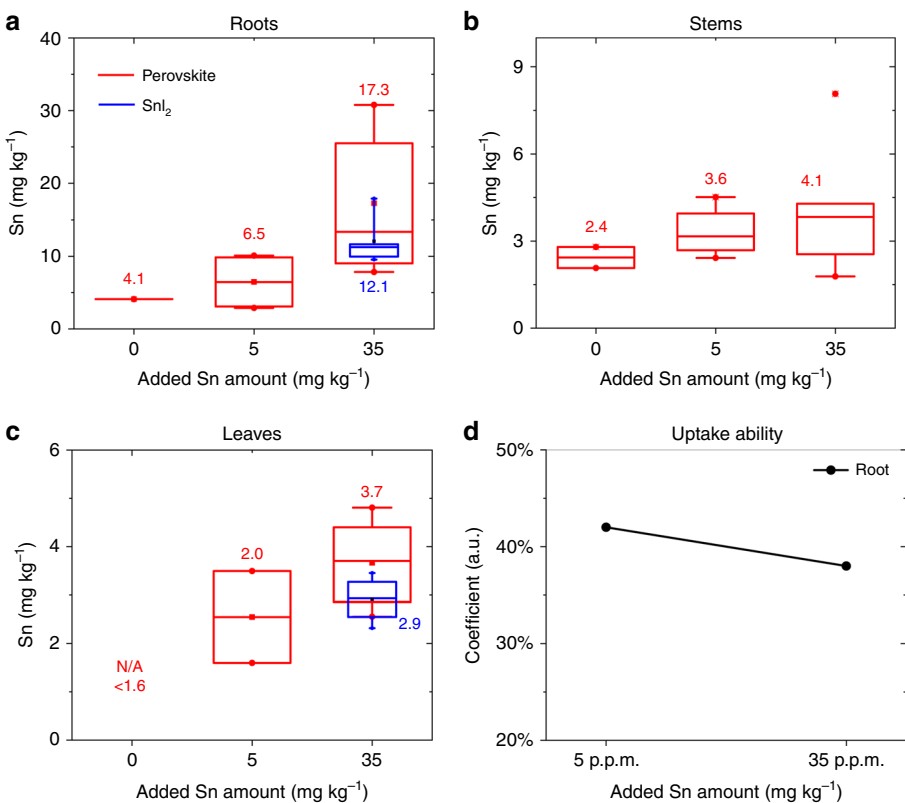

**Fig. 3 Tin concentration in different parts of mint plants. a** Roots, **b** stems, and **c** leaves. The tin uptake ability ($\eta$, defined in Eq. 1) is reported in **d**. The maximum, minimum, and mean value of each distribution are reported in **a**–**c**.

tin perovskite inhibits the seed germination and early seedling (see Supplementary Fig. 5).

## Discussion

We provide a data set that uses plants to address the environmental impact of the halide perovskites. We measure the lead uptake ability or bioavailability by plants in halide perovskite-contaminated soil. We find that the lead from halide perovskite is more dangerous than other sources of the lead contamination already present in the ground as it is ten times more bioavailable. We highlight that the environmental impact of lead perovskite cannot be estimated on a linear scale of effect versus concentration, as it was tacitly assumed up to now. We also prove that tin perovskites, which are often proposed as a safer option, are significantly less bioavailable than lead-based perovskite. Our studies demonstrate the urge of a systematic screening of the environmental impact of different perovskite compositions to drive the application of these advanced class of materials towards a sustainable market.

## Methods

The detailed treatment of soil samples is shown in Supplementary Note 4. Thirty days old *Mentha spicata* plants were purchased from Qingdao Baicaoxiang Fangxiang plants Co. Ltd, and then grown in the hood for 20 days. Five plants were grown each in 100 g of soils. During the growth stages, the plants were watered with 50 mL of water every 2 days. After growth, fresh mint plants were separated from the soil. They were rinsed with water and ultra-sonicated for 2 min to remove the adhering soil particles and dust. Then mint roots, stems, and leaves were separated and dried at 60 °C in an oven for 48 h. Then the samples were chopped into pieces, and a representative sample of 0.1 g was taken for metal concentration measurement. The metal concentration was measured by inductively coupled plasma mass spectrometry (ICP-MS, Agilent Technology 7700 series). CCIC-FUJIAN MINERALS INSPECTION&TESTING CO., LTD certificated the metal concentration.

## Data availability

The data that support the findings of this study are available from the corresponding authors upon reasonable request.

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

## Acknowledgements

A. Abate thanks the National Natural Science Foundation of China (21750110442). J.Lü acknowledges the International Science and Technology Cooperation and Exchange Project of Fujian Agriculture and Forestry University (No. KXGH17010) for funding. J.Li. is supported by the International Postdoctoral Exchange Fellowship Program between Helmholtz-Zentrum für Materialien und Energie GmbH, OCPC and Fuzhou University. H.L. Cao thanks the financial support from the Fujian Agriculture and Forestry University Program for Distinguished Young Scholar (Grant NO. xjq201813).

## Author contributions

J.Li. performed the experimental work about tin and data analysis. H.C. performed the experimental work about lead. W.J. performed the ICP experiments. Q.W. drafted the manuscript. The project was conceived, planned, and supervised by M.W., I.C., J.Lü and A.A. All authors discussed the results and provided feedback on the manuscript.

## Competing interests

The authors declare no competing interests.
