## [Peer Review File · Nature Communications]

Reviewers' comments:

Reviewer #1 (Remarks to the Author):

In this paper, the authors describe how (1) the uptake of Pb by plants (mint) is supra-linear with concentration of Pb in the soil (i.e. the uptake increases faster than does the Pb concentration. (2) the uptake depends on the source of Pb, suggested due to the effect of organic cations on the pH of the soil. (3) Uptake of Sn is less than that of Pb for the same concentrations in soil. This is an important study as it shows that the environmental effects of Pb cannot simply be estimated on a linear scale of effect vs. concentration as was tacitly assumed up to now. I recommend publication after a number of relatively minor issues have been taken care of.

P3, end of para. 1. 'This study provides the first set of experimental data to assess the environmental risk of halide perovskites.'

It is not the first. See e.g. ref. 10 that gives experimental data. It IS the first (that I know of) that considers Pb uptake by plants and this is an important distinction from earlier studies.

P3, line 2 of para. 2. 'We measured the natural lead concentration of soil around 36 mg/kg..' The Pb concentration in soil can vary widely, in particular between urban and not-urban locations. The authors should describe the location(s) of the soil they used.

P5, lines 6/7. samples with 250 mg/kg of lead, since this is the limit imposed for agricultural soil¹³. Is this a global limit, a Chinese one, or what? More detail needed here.

P8, legend to fig. 3. 'Thus we do not show the data about stems since it is hard to distinguish roots and stems.'

I assume what the authors meant was that they do not show the data for Sn¹² about stems...'

p8, 2/3 lines after the fig. legend. 'tin-based perovskites are one of the most exciting possibility¹⁹ since Sn²⁺ can rapidly oxidise to toxicologically inactive Sn⁴⁺ compounds..'

While this is true, Sn²⁺ is also less toxic (to humans and animals in general; it can be toxic to aquatic life) than Pb²⁺. as presently written it gives the impression that the reason Sn is better than Pb is solely due to the oxidation.

P8, line 4. 'To validate the hypothesis that tin is less toxic than lead...,'

This does not need validation. What the authors show is that Sn uptake is less than that of Pb.

P10, In the Methods, two important things should be elaborated.

1. It was written that five plants were each grown in 100 g of soil. However, the depth of the soil and the drainage (if any) should also be given. The drainage system is of particular importance. For example, since the soil was thoroughly dried and then the Pb-contaminated water added (50 ml to 100 g soil, that lead will stay in the soil. I assume the lead was added only to the first irrigation and not to subsequent ones. This is crucial and should be explicitly stated.

2. The water used for irrigation should be described. Was it tap water, distilled water, etc.? How much lead (if any) was in the water?

Grammatical corrections.

P6, 8 lines from bottom. change 'non-from perovskite' to 'not from perovskite'.

P7, line 3. Change 'the few plants survived' to 'the few plants that survived'.

P7, line 6. Change 'to describe the mints uptake' to 'describe the mint plants uptake'. similarly for p8, legend to the fig. and p9. 2nd para.

P7, line 4 after eq. 1. Change 'improves' to 'increases'.

P7, 2 lines from the end. Change 'influence' to 'influences'.

P8, 3 lines after the fig. legend. Change 'possibility' to 'possibilities'.

Reviewer #2 (Remarks to the Author):

The manuscript reports on an investigation regarding the toxicity issue related to perovskite solar cells. Given that all the well performing perovskite solar cells contain Pb, or to a lesser extent Sn, leakage of either of these in the environment is certainly undesirable if not unacceptable, so the potential associated impact must be well understood. So far, only a few studies have addressed this topic, despite its relevance. The current study looks into the lead content of mint plants that have been grown in soil contaminated with Pb from $\text{MAPbI}_3/\text{PbI}_2$ and $\text{SnPbI}_3/\text{SnI}_2$. The study is interesting and relevant to the entire halide perovskite community, also outside photovoltaics. Regrettably, there are many problems with the manuscript and many open questions. A number of concerns are listed below:

There is a lot of unclarity regarding the experiment itself:

§ How is the perovskite added to the soil, and how is it processed to make sure it is homogeneously distributed? After all, soil itself is not exactly a homogeneous mixture, and the amount of added contaminant is minute.

§ How sure is it that the Pb already in the soil is homogeneously distributed?

§ The plants were rinsed with water and sonicated for two minutes to remove soil particles and dust. How can it be confirmed that this method is effective? Even a small residue can lead to skewed results, especially given that the samples were chopped into pieces, and from those pieces only 0.1g was taken for analysis.

§§ How is this chopping performed exactly, and is this an established method? Also, this does not factor in that Pb does not necessarily have to accumulate homogeneously in the root, and/or stems and/or leaves. Has this been checked?

§ In the study, 30 day old plants have been used for a 20 day intoxication, before the plants are harvested and analyzed. This completely negates the very important first 30 days: the germination and early development of the plants. To be fully representative, it should have been investigated what happens right from the moment when seeds are sown. As an analogy, in reference 9 it has been shown that the hatching of zebrafish embryos is heavily harmed by Sn compounds, more than Pb, so in that study Sn was found more toxic as a net result, but not as a result of heavy metal intoxication. Since Sn is known for its ability to acidify, in case of the plants this may lead to the inability of seeds to germinate in the first place. The current analysis does not provide enough information to identify a mechanism or make important distinctions as to why the Sn case is different than Pb. Also, to answer any of these questions, it should at least be known as what compound Pb and Sn are available in the soil as it is, but this is not even touched upon.

§ Is there any way to obtain a pure soil sample, (without ANY Pb in it). That would be the actual ideal reference point.

§ The given equation for the uptake ability is only valid for the cases where there is contamination from perovskite, but not for the presence of Pb already in the soil from other sources than perovskite, which makes it quite unclear what can be learnt from it.

§ on page 8, it is stated "...since Sn can rapidly oxidise to toxicologically inactive Sn compounds if dispersed in the environment" Where does the information about less toxic Sn⁴⁺ species come from? Is there a reference to support this?

Some additional aspects that are unclear, incomplete, incorrect or inconsistent:

§ Given the nature of the experiment, it would be plausible to expect relatively large fluctuations in results. Unfortunately, it is not mentioned how many plants have been investigated for each condition. It is also not clear if the markers in Fig. 2 and 3 are individual data points, or statistical indicators of the box plots. If there would be only two or three plants per condition, it would cast a lot of uncertainty. I would strongly recommend adding all the data points in the plot, or at least mention the number of plants per condition explicitly.

§ on page 8 it reads: "Not surprisingly, we found that the organic cation has a substantial impact on the lead uptake ability of mint. The data reported in Fig. 2 show that the lead content in mint is similar or even lower than in plants grown in natural soil." This last sentence is very unclear. What I think it meant is "The lead content in mint grown in soil contaminated with perovskite (PbI₂+MAI) is similar or even lower than in plants grown in natural soil." If that is not what is meant, the authors should improve their wording and make things clear. If that is what is meant, it is incorrect, based on Figure 2, where the red "perovskite" data is always higher than the blue "PbI₂" data.

§§ On the topic of this data, it is incomplete. Why is there no PbI₂ data for the 0 and 35mg/kg cases for Pb, and for the 0 and 5mg/kg cases for Sn?

§ The abstract mentions "standard lead contaminating soil". I assume what is meant is "standard lead contaminatED soil", but more importantly, this is not an appropriate term to use in the abstract of a manuscript dealing with such a sensitive topic. It suggests that soil is somehow contaminated by default, or that it is there naturally, which is not at all the case. Related to this, on page 3, it reads: "We measured the natural lead concentration of soil around 36 mg/kg,...". The term "natural lead concentration" is deceptive. The lead concentration in the soil is non-zero because humans started to mine lead, with soil contamination as a result (reference should be made to: Babayigit et al. (10.1557/mre.2017.17)). Otherwise it would have never moved beyond the earth's crust. Using the term "natural lead concentration" also creates the impression that, to some standard, a small increase in the concentration is unimportant, while it is in fact yet another undesirable increase on top of the unfortunate concentration mankind has already induced throughout history. The lead concentration could be termed as "effective", but definitely not "natural". All other instances of "natural" in the manuscript should also be removed for that matter. The context of the paragraph actually does not make it clear to me whether the authors are not aware of this or they simply have not expressed it explicitly. Whichever it may be, it creates a strong bias from the start, and it is important to refrain from the use of any phrase containing "natural lead" altogether, given the sensitivity of the subject at hand.

§ The authors fail to mention that for certain meat products the WHO maximum tolerable tin concentration is only 50mg/kg

(http://www.fao.org/tempref/codex/Meetings/CCCF/CCCF5/cf05_INF.pdf)

§ To the best of my knowledge, this is the first study that uses plants to evaluate the toxicity of such halide perovskites or degradation products thereof. On the other hand, the authors do make overstated claims, such as:

p2: "this work provides the first experimental evidence that we need to treat PSCs with exceptional care".

p3: "This study provides the first set of experimental data to assess the environmental risk of halide perovskites "

p9: "we provided the first data set addressing the environmental impact of the lead-based halide perovskites"

Wherever and whenever this work could be published, these statements have to be removed or altered, as reference 9 and Benmessaoud et al. (10.1039/C5TX00303B) have also ventured in this direction, albeit with different approaches.

Finally, the authors conclude "We also proved that tin perovskites, which are often proposed as a safe alternative, are effectively safer than lead-based perovskite". This statement lacks any form of nuance. I consider it highly inappropriate to draw this conclusion based on one set of experiments by one research unit, on one type of organism. At best the results give some indication that this could be the case, but a vast amount of additional work needs to be done to come to the overall conclusion as stated in the manuscript. Only one type of plant is used in one type of experiment, and while this is a necessary start, it is by no means an inclusive study. With this, and certainly considering the manuscript's many flaws I addressed in this report, the conclusion of the manuscript is far too generalized and oversells the work. The study reports observations but fails to reveal any mechanism. It is interesting but for now raises more questions than it answers. As such, it is preliminary and therefore not suited for Nature Communications.

Reviewer #1 (Remarks to the Author)

In this paper, the authors describe how (1) the uptake of Pb by plants (mint) is supra-linear with concentration of Pb in the soil (i.e. the uptake increases faster than does the Pb concentration. (2) the uptake depends on the source of Pb, suggested due to the effect of organic cations on the pH of the soil. (3) Uptake of Sn is less than that of Pb for the same concentrations in soil. This is an important study as it shows that the environmental effects of Pb cannot simply be estimated on a linear scale of effect vs. concentration as was tacitly assumed up to now.

I recommend publication after a number of relatively minor issues have been taken care of.

P3, end of para. 1. 'This study provides the first set of experimental data to assess the environmental risk of halide perovskites.'

It is not the first. See e.g. ref. 10 that gives experimental data. It IS the first (that I know of) that considers Pb uptake by plants and this is an important distinction from earlier studies.

Response 1-1:

Thank you for noting this. We agree with the reviewer, in the ref. 10 the authors investigated for the first time how much lead can leak from perovskite modules by exposing them to the rain after damaging the module. In our study, we reported for the first time the use of plants to assess the environmental risk of halide perovskite leakage. We have modified the text into the manuscript as follow:

P3, end of para.1: This study provides for the first time the use of plants to assess the environmental risk of halide perovskite.

P3, line 2 of para. 2. 'We measured the natural lead concentration of soil around 36 mg/kg.'

The Pb concentration in soil can vary widely, in particular between urban and not-urban locations. The authors should describe the location(s) of the soil they used.

Response 1-2:

In this study, we collected the samples from the ground in Fujian Agriculture and Forestry University, China (latitude 26.084, longitude 119.238). We added more information about the soils used in this study, as shown in SI para. 1:

ESI, P1, para 1: The planting experiment were carried out according to a standard toxicity testing method. The soil was taken from the ground (managed and protected by University) in Fujian Agriculture and Forestry University, China (latitude 26.084, longitude 119.238). The soil samples were firstly air-dried about ten days to remove the moisture content and the residual plants visible to naked eyes; then grind and filtered (2-mm-filter) to remove the sundries (as shown in Figure S1). We measured the Pb concentration from five soil samples (Table S1): which were 37.7, 38.3, 35.2, 32.2, and 38.1 mg/kg, respectively, and the mean value was 36.3 mg/kg.

Figure S1. The treatments of soil: (a) air-dry, (b) grind, and (c) filter

Table S1: Pb concentration in natural soil

Sample	01	02	03	04	05	Mean
concentration (mg/kg)	37.7	38.3	35.2	32.2	38.1	36.3

Meanwhile, we modified the description in the manuscript as:

- **P4 Figure 1 legend:** The inset shows the photo of mint plants grown in perovskite-contaminated soil within the campus of Fujian Agriculture and Forestry University, China (latitude 26.084, longitude 119.238).
- **P10, line 1:** The detailed treatment of soil samples is shown at experiment details part 1 in supporting information (SI).
- **P3, para. 2, line 2:** We measured the lead concentration in natural soil around 36 mg/kg (Table S1), which is in line with the value reported in the literature.

We verified that 36mg/Kg well represent the lead concentration in agricultural soil for several countries world wild.

P5, lines 6/7. samples with 250 mg/kg of lead, since this is the limit imposed for agricultural soil¹³. Is this a global limit, a Chinese one, or what? More detail needed here.

Response 1-3:

250 mg/kg is the maximum lead concentration in soil allowed for agriculture in China. Over the world, the maximum lead concentration allowed change significantly ^[a], for example:

Country	China	Australia	Canada	France	Germany	Japan	Netherlands	UK	USA
Values (ppm)	250	100	140	400	400	150	530	450	400

^a: CHEN, S.-b., et al., Overview of current criteria for heavy metals and its hint for the revision of soil environmental quality standards in China. Journal of integrative agriculture, 2018. 17(4): p. 765-774.

We note that the Chinese regulation is one of the most restrictive.

We have provided more details on the manuscript by adding:

- **P5, para.1, line 6/7:** Finally, we prepared a collection of samples with 250 mg/kg of lead, since this is the limit imposed for agricultural soil in China.
- **ESI, P1 para. 1:** Over the world, the maximum lead concentration in land for agriculture are varied in countries, as shown in Table S2:

Table S2: Environmental quality standards for lead in soil of different countries ^a

Country	China	Australia	Canada	France	Germany	Japan	Netherlands	UK	USA
Values (ppm)	250	100	140	400	400	150	530	450	400

^a: CHEN, S.-b., et al., Overview of current criteria for heavy metals and its hint for the revision of soil environmental quality standards in China. Journal of integrative agriculture, 2018. 17(4): p. 765-774.

P8, legend to fig. 3. 'Thus we do not show the data about stems since it is hard to distinguish roots and stems.'

I assume what the authors meant was that they do not show the data for SnI2 about stems...'

Response 1-4:

Yes, that is what we mean. We have modified our sentence as:

Legend to Fig. 3: Thus, we do not show the data for SnI₂ about stems since it is hard to distinguish roots and stems.'

p8, 2/3 lines after the fig. legend. 'tin-based perovskites are one of the most exciting possibility¹⁹ since Sn²⁺ can rapidly oxidise to toxicologically inactive Sn⁴⁺ compounds..'

While this is true, Sn²⁺ is also less toxic (to humans and animals in general; it can be toxic to aquatic life) than Pb²⁺. as presently written it gives the impression that the reason Sn is better than Pb is solely due to the oxidation.

Response 1-5:

We thank you for noting this, and we have modified our sentence as:

• **P8, para. 1, line 2/3/4:** tin-based perovskites are one of the most exciting possibilities ^[23], since Sn is less toxic than lead to humans and animals in general (it can be toxic to aquatic life) ^[24]; moreover, if Sn²⁺ compounds were dispersed in the environment, it can rapidly be oxidized to stable oxygenated Sn⁴⁺ compounds with a low solubility in water.

P8, line 4. 'To validate the hypothesis that tin is less toxic than lead...,'

This does not need validation. What the authors show is that Sn uptake is less than that of Pb.

Response 1-6:

We thank you for noting this, we have modified our sentence as:

P8, para 1. line 5/6: To investigate the environmental impact of tin perovskites leakage into the soil, we repeated the same set of experiments reported for lead-based perovskite

P10, In the Methods, two important things should be elaborated.

1. It was written that five plants were each grown in 100 g of soil. However, the depth of the soil and the drainage (if any) should also be given. The drainage system is of particular importance. For example, since the soil was thoroughly dried and then the Pb-contaminated water added (50 ml to 100 g soil, that lead will stay in the soil. I assume the lead was added only to the first irrigation and not to subsequent ones. This is crucial and should be explicitly stated.

2. The water used for irrigation should be described. Was it tap water, distilled water, etc.? How much lead (if any) was in the water?

Response 1-7:

We thank you for noting this; 1) The depth of the soil is around 8 cm. there is drainage in the mint grow flowerpot. During the experiment, there is no water leakage from the drainage system, while the soil is humid. 2) The water used for irrigation is de-ionised water (>18 M). To give a more detailed explanation about the experiment methods, we have added the following descriptions in ESI:

ESI, P1 and P2:

1. Soil treatment

The planting experiment was carried out according to a standard toxicity experiment method. The soils were taken from the group based (managed and protected by the university) in Fujian Agriculture and Forestry University, China (latitude 26.084, longitude 119.238). The Pb/Sn exist in natural soil in several speciations: exchangeable, carbonate bounded, oxides, organic and residual form. The soil samples were firstly air-dried about ten

days to remove the moisture content; then the soil samples were grinded and filtered (2-mm-filter) to remove the sundries, as shown in Figure S1. We measured the Pb concentration from five soil samples: which were 37.7, 38.3, 35.2, 32.2, and 38.1 mg/kg, respectively, and the mean value was 36.3 mg/kg (Table S1). In China, where the more significant production of lead is concentrated, the agricultural regulation tolerates a lead content into the soil up to 250 mg/Kg; and over the world, the maximum lead concentration in land for agriculture are varied in countries, as shown in Table S2:

Figure S1. The treatments of soil: (a) air-dry, (b) grind, and (c) filter

Table S1: Pb concentration in natural soil in this experiment

Sample	01	02	03	04	05	Mean
concentration (mg/kg)	37.7	38.3	35.2	32.2	38.1	36.3

Table S2: Environmental quality standards for lead in soil of different countries a

Country	China	Australia	Canada	France	Germany	Japan	Netherlands	UK	USA
Values (ppm)	250	100	140	400	400	150	530	450	400

The fundamental chemical reaction for perovskites (for example, MAPbI₃) is reversible, as follows:

In the presence of oxygen, water, and moisture, etc., the chemical equilibrium can drive the reaction in the negative direction, leading to the decomposition of MAPbI₃ into MAI and PbI₂ [a-d]. Thus, for simplicity, we added MAI and PbI₂ power instead of perovskite (MAPbI₃) in soil. In this study, both MAI+PbI₂ and PbI₂ were added into an independent flowerpot (as

shown in Figure S2). Then, the soils were stirred for 50 times with a spatula. We measured the lead concentration in 5-ppm lead perovskite contaminated soil as an example; the data were shown in Table S3. Thus, the additives (MAI+PbI₂ or PbI₂) were supposed to be homogeneously distributed in soil.

Figure S2. The photo of soil in the independent flowerpot

Table S3: Pb concentration in 5-ppm lead perovskite contaminated soil

Sample	01	02	03	04	05	Mean
concentration (mg/kg)	40.8	40.1	41.3	39.1	41.9	40.6

^a: CHEN, S.-b., et al., Overview of current criteria for heavy metals and its hint for the revision of soil environmental quality standards in China. *Journal of integrative agriculture*, 2018. 17(4): p. 765-774.

2. Mint plants grow

Thirty days old *Mentha Spicata* plants, which born with cutting propagation technique and produced with water culture method, were purchased from Qingdao Baicaoxiang Fangxiang plants Co. Ltd. We have measured the lead/tin concentration in the 30 days old mint, and we found the tin concentration is shallow (tin < 1.6 mg/kg, Tables S2 and Table S3) thus it is not necessary to start the experiment from mint seed sow. These mints were grown in the 8 cm thick soil for 20 days. During the growth, the root of the mint was fully immersed in soil (as shown in Figure S3).

Figure S3. The photo of mint; left: born with cutting propagation technique and grown with water culture method; middle: the mint used for the experiment; and right: mint in the flowerpot.

Table S4. Lead concentration in the 30-days old mint used for the experiment

Sample	01	02	03	04	05	Mean
roots (mg/kg)	1.8	0.4	0.3	0.9	2.0	1.1
stems (mg/kg)	1.0	0.9	1.7	/	0.5	0.5
Leaves (mg/kg)	0.7	0.4	0.2	0.4	0.4	0.4

Table S5. Tin concentration in 30-days old mint used for the experiment

Sample	01	02	03	04	05	Mean
roots (mg/kg)	<1.6	<1.6	<1.6	<1.6	<1.6	<1.6
stems (mg/kg)	1.6	<1.6	<1.6	<1.6	<1.6	<1.6
Leaves	<1.6	<1.6	<1.6	<1.6	<1.6	<1.6

(mg/kg)						
---------	--	--	--	--	--	--

* The detect limitation for tin concentration is 1.6 mg/kg.

Five plants were grown each in 100 g of soils. During the growth stages, the plants were watered with 10 mL of de-ionised water (>18 M, without any metal cations) every two days. There is no water leakage from the drainage system while the soil is humid during the experiment. During the growth, the room temperature was set as 20 °C. Every day, the plants were illuminated with 104 lux light for 16 hours and other 8 hours in the dark. After harvest, they were rinsed with de-ionised water and ultra-sonicated for 2 min to remove the adhering soil particles and dust. Then mint roots, stems and leaves were separated and dried at 60 °C in an oven for 48 hours. Then the samples were chopped into pieces, and a representative sample of 0.1 g was taken for metal concentration measurement by inductively coupled plasma mass spectrometry (ICP-MS, Agilent Technology 7700 series).

Figure S4. The photo of part mint plants experiment.

Grammatical corrections.

P6, 8 lines from bottom. change 'non-from perovskite' to 'not from perovskite'.

P7, line 3. Change 'the few plants survived' to 'the few plants that survived'.

P7, line 6. Change 'to describe the mints uptake' to 'describe the mint plants uptake'. similarly for p8, legend to the fig. and p9. 2nd para.

P7, line 4 after eq. 1. Change 'improves' to 'increases'.

P7, 2 lines from the end. Change 'influence' to 'influences'.

P8, 3 lines after the fig. legend. Change 'possibility' to 'possibilities'.

Response 1-8:

We thank you for noting these grammatical mistakes; we have now corrected all these grammatical mistakes:

- **P6, line 9 from bottom:** 'non-from perovskite' to 'not from perovskite'
- **P7, line 3:** 'the few plants survived' to 'the few plants that survived'
- **P7, line 7:** 'describe the mints uptake' to 'describe the mint plants uptake'
- **P8 legend:** 'the mints are 'to 'the mints plants are'
- **P9, para.3, line 2:** 'mint in halide perovskite' to 'mint plants in halide perovskite'
- **P7 line 4** after eq. 1: 'Improves' to 'increases'
- **P7 line 2** from the end: 'Influence' to 'influences'
- **P8 line 3:** 'possibility' to 'possibilities'

Reviewer #2 (Remarks to the Author)

The manuscript reports on an investigation regarding the toxicity issue related to perovskite solar cells. Given that all the well performing perovskite solar cells contain Pb, or to a lesser extent Sn, leakage of either of these in the environment is certainly undesirable if not unacceptable, so the potential associated impact must be well understood. So far, only a few studies have addressed this topic, despite its relevance. The current study looks into the lead content of mint plants that have been grown in soil contaminated with Pb from $\text{MAPbI}_3/\text{PbI}_2$ and $\text{SnPbI}_3/\text{SnI}_2$. The study is interesting and relevant to the entire halide perovskite community, also outside photovoltaics. Regrettably, there are many problems with the manuscript and many open questions. A number of concerns are listed below:

There is a lot of unclarity regarding the experiment itself:

§ How is the perovskite added to the soil, and how is it processed to make sure it is homogeneously distributed? After all, soil itself is not exactly a homogeneous mixture, and the amount of added contaminant is minute.

Response 2-1:

We thank the reviewer for noting this; we provided more details about the soil treatment and mint growth, as shown in the ESI part 1 and part 2.

ESI, Part 1 and 2:

1. Soil treatment

The planting experiment was carried out according to a standard toxicity experiment method. The soils were taken from ground based (managed and protected by the university) in Fujian Agriculture and Forestry University, China (latitude 26.084, longitude 119.238). The Pb/Sn exist in natural soil in several speciations: exchangeable form, carbonate bounded form, oxides form, organic form, and residual form. The soil samples were firstly air-dried about ten days to remove the moisture content; then the soil samples were grinded and filtered (2-mm-filter) to remove the sundries, as shown in Figure S1. We measured the Pb concentration from five soil samples: which were 37.7, 38.3, 35.2, 32.2, and 38.1 mg/kg, respectively, and the mean value was 36.3 mg/kg (Table S1). In China, where the more significant production of lead is concentrated, the agricultural regulation tolerates a lead content into the soil up to 250 mg/Kg; and over the world, the maximum lead concentration in land for agriculture are varied in countries, as shown in Table S2:

Figure S1. The treatments of soil: (a) air-dry, (b) grind, and (c) filter

Table S1: Pb concentration in natural soil in this experiment

Sample	01	02	03	04	05	Mean
concentration (mg/kg)	37.7	38.3	35.2	32.2	38.1	36.3

Table S2: Environmental quality standards for lead in soil of different countries ^a

Country	China	Australia	Canada	France	Germany	Japan	Netherlands	UK	USA
Values (ppm)	250	100	140	400	400	150	530	450	400

The fundamental chemical reaction for perovskites (for example, MAPbI₃) is reversible, as follows:

In the presence of oxygen, water, and moisture, etc., the chemical equilibrium can drive the reaction in the negative direction, leading to the decomposition of MAPbI₃ into MAI and PbI₂ [b-e]. Thus, for simplicity, we added MAI and PbI₂ power instead of perovskite (MAPbI₃) in soil. In this study, both MAI+PbI₂ and PbI₂ were added into an independent flowerpot (as shown in Figure S2). Then, the soils were stirred for 50 times with a spatula. We measured the lead concentration in 5-ppm lead perovskite contaminated soil as an example; the data were shown in Table S3. Thus, the additives (MAI+PbI₂ or PbI₂) were supposed to be homogeneously distributed in soil.

Figure S2. The photo of soil in the independent flowerpot

Table S3: Pb concentration in 5-ppm lead perovskite contaminated soil

Sample	01	02	03	04	05	Mean
concentration (mg/kg)	40.8	40.1	41.3	39.1	41.9	40.6

Reference:

- ^a: CHEN, S.-b., et al., Overview of current criteria for heavy metals and its hint for the revision of soil environmental quality standards in China. *Journal of integrative agriculture*, 2018. 17(4): p. 765-774.
- ^b. Conings, B. et al. Intrinsic Thermal Instability of Methylammonium Lead Trihalide Perovskite. *Advanced Energy Materials* 5, 1500477, (2015).
- ^c. Han, Y. et al. Degradation observations of encapsulated planar CH₃NH₃PbI₃ perovskite solar cells at high temperatures and humidity. *Journal of Materials Chemistry A* 3, 8139-8147, (2015).
- ^d. Shahbazi, M. & Wang, H. Progress in research on the stability of organometal perovskite solar cells. *Solar Energy* 123, 74-87 (2016).
- ^e. Yang, J. et al. Investigation of CH₃NH₃PbI₃ degradation rates and mechanisms in controlled humidity environments using in situ techniques. *Acs Nano* 9, 1955-1963 (2015).

2. Mint plants grow

Thirty days old *Mentha Spicata* plants, which born with cutting propagation technique and produced with water culture method, were purchased from Qingdao Baicaoxiang Fangxiang plants Co. Ltd. We have measured the lead/tin concentration in the 30 days old mint, and we found the tin concentration is shallow (tin < 1.6 mg/kg, Tables S2 and Table S3) thus it is not necessary to start the experiment from mint seed sow. These mints were grown in the 8 cm thick soil for 20 days. During the growth, the mint root was fully immersed in soil (as shown in Figure S3).

Figure S3. The photo of mint; left: born with cutting propagation technique and grown with water culture method; middle: the mint used for the experiment; and right: mint in the flowerpot.

Table S4. Lead concentration in the 30-days old mint used for the experiment

Sample	01	02	03	04	05	Mean
roots (mg/kg)	1.8	0.4	0.3	0.9	2.0	1.1
stems (mg/kg)	1.0	0.9	1.7	/	0.5	0.5
Leaves (mg/kg)	0.7	0.4	0.2	0.4	0.4	0.4

Table S5. Tin concentration in 30-days old mint used for the experiment

Sample	01	02	03	04	05	Mean
roots (mg/kg)	<1.6	<1.6	<1.6	<1.6	<1.6	<1.6
stems (mg/kg)	1.6	<1.6	<1.6	<1.6	<1.6	<1.6
Leaves	<1.6	<1.6	<1.6	<1.6	<1.6	<1.6

(mg/kg)						
---------	--	--	--	--	--	--

* The detect limitation for tin concentration is 1.6 mg/kg.

Five plants were grown each in 100 g of soils. During the growth stages, the plants were watered with 10 mL of de-ionised water (>18 M, without any metal cations) every two days. There is no water leakage from the drainage system while the soil is humid during the experiment. During the growth, the room temperature was set as 20 °C. Every day, the plants were illuminated with 104 lux light for 16 hours and other 8 hours in the dark. After harvest, they were rinsed with de-ionised water and ultra-sonicated for 2 min to remove the adhering soil particles and dust. Then mint roots, stems and leaves were separated and dried at 60 °C in an oven for 48 hours. Then the samples were chopped into pieces, and a representative sample of 0.1 g was taken for metal concentration measurement by inductively coupled plasma mass spectrometry (ICP-MS, Agilent Technology 7700 series).

Figure S4. The photo of part mint plants experiment.

§ How sure is it that the Pb already in the soil is homogeneously distributed?

Response 2-2:

We randomly measured the Pb already in the natural soil samples (as shown in table 1); which were 37.7, 38.3, 35.2, 32.2, and 38.1mg/kg; the mean concentration is 36.3 mg/kg.(as shown in **table S1**). Thus, we conclude that the lead concentration in natural soil is uniformly distributed.

§ The plants were rinsed with water and sonicated for two minutes to remove soil particles and dust. How can it be confirmed that this method is effective? Even a small residue can lead to skewed results, especially given that the samples were chopped into pieces, and from those pieces only 0.1g was taken for analysis.

Response 2-3:

We agree with the reviewer, the plants, especially the roots, are complicated to clean; and a small residue can lead to skewed results. However, we repeated several times the same procedure, and we found consistent results. For this study, we adopted an established method for substantial metal uptake by plants: The plants were rinsed with de-ionised water and high two min to remove the adhering soil particles and dust. This procedure was confirmed by Ref^[e].

Refer:

[e] Cuske, M. et al. Ultrasonic cleaning of plant roots in their preparation for analysis on heavy metals. *Zeszyty Naukowe. Inżynieria Środowiska/Uniwersytet Zielonogórski*, 25-32 (2014).

§§ How is this chopping performed exactly, and is this an established method? Also, this does not factor in that Pb does not necessarily have to accumulate homogeneously in the root, and/or stems and/or leaves. Has this been checked?

Response 2-4: The methods we used is an established preparation method to characterise the heavy metal concentration in plants (for example, in ref. [f, g]). Based on this comment, we would like to give more description of experimental details in ESI.

ESI P5, Para 1, line 7: After harvest and cleaning of plants, the mint roots, stems and leaves were separated and dried at 60 °C in an oven for 48 hours. Then these plants were chopped into several small pieces (<0.2mm). A representative sample of 0.1 g was taken for metal concentration measurement by inductively coupled plasma mass spectrometry.

Ref.

f. Arias, J.A., et al., Plant growth and metal distribution in tissues of *Prosopis juliflora-velutina* grown on chromium contaminated soil in the presence of *Glomus deserticola*. *Environmental science & technology*, 2010. 44(19): p. 7272-7279.

g. Hoseini, P.S., et al., Ability of phytoremediation for the absorption of strontium and caesium from soils using *Cannabis sativa*. *International Journal of Environmental Health Engineering*, 2012. 1(1): p. 17.

We also agree with the reviewer that Pb may not accumulate homogeneously. However, the concentration is so low that it was not possible to study the distribution of lead into the different part of the plants. The best we could make was to separate roots from stems and leaves to mimic the most common pieces of the plants that can be used as food.

§ In the study, 30-day old plants have been used for a 20-day intoxication, before the plants are harvested and analyzed. This completely negates the very important first 30 days: the germination and early development of the plants. To be fully representative, it should have been investigated what happens right from the moment when seeds are sown. As an analogy, in reference 9 it has been shown that the hatching of zebrafish embryos is heavily harmed by Sn compounds, more than Pb, so in that study Sn was found more toxic as a net result, but not as a result of heavy metal intoxication. Since Sn is known for its ability to acidify, in case of the plants this may lead to the inability of seeds to germinate in the first place. The current analysis does not provide enough information to identify a mechanism or make important distinctions as to why the Sn case is different than Pb. Also, to answer any of these questions, it should at least be known as what compound Pb and Sn are available in the soil as it is, but this is not even touched upon.

Response 2-5:

a) The mint plants were born with cutting propagation technique and grown with water culture method, instead of from seeds sown. We measured the lead/tin concentration in the 30 days old mint, and we found the tin concentration are shallow (Pb/Sn < 1.6 mg/kg, Tables S4 and Table S5).

Table S4. Lead concentration in 30-day old mint used for the experiment

Sample	01	02	03	04	05	Mean
roots (mg/kg)	1.8	0.4	0.3	0.9	2.0	1.1
stems (mg/kg)	1.0	0.9	1.7	/	0.5	0.5
Leaves (mg/kg)	0.7	0.4	0.2	0.4	0.4	0.4

Table S5. Tin concentration in 30-day old mint used for the experiment

Sample	01	02	03	04	05	Mean
roots (mg/kg)	<1.6	<1.6	<1.6	<1.6	<1.6	<1.6
stems (mg/kg)	1.6	<1.6	<1.6	<1.6	<1.6	<1.6
Leaves (mg/kg)	<1.6	<1.6	<1.6	<1.6	<1.6	<1.6

* The detect limitation for tin concentration is 1.6 mg/kg.

We perform additional experiments based on *Brassica campestris* (Cabbage) growth from seeds sown to investigate the influence of Pb/Sn leakage at the early development stage of the plants. We found that Pb/Sn perovskite and PbI_2/SnI_2 can reduce the ability of seeds to germinate and reduce plant growth. Under the 35 and 250 ppm Pb/Sn concentration, the seed germination and early seedling growth were reduced compared to control. Moreover,

the seeds grow more slowly in Sn cations contaminated soil. The effect of Sn could be due to the higher ability of Sn to acidify than Pb, which affect seed growth ^[a].

Figure S5. The photo of cabbage seed germination and seedling growth.

b) The Pb/Sn could exist in natural soil in several speciations: exchangeable form, carbonate bounded form, oxides form, organic form, and residual form. Only parts of Pb/Sn could be uptake by plants from natural soil. In our study, we investigate the Pb/Sn cations uptake by plants from perovskite-contaminated soil. In theory, all (or most of) the Pb/Sn cations could be uptake by plants. This could explain why the mint plants can uptake much more Pb/Sn from perovskite-contaminated soil than from natural land.

The ability of plants to uptake heavy metals from the ground, i.e. the bioavailability of the metals, is influenced by pH, the metal concentration, metal cation exchange capability between roots and soil, and many other factors. Unfortunately, based on this work, we could not identify a mechanism to why the Sn case is different than Pb.

Based on this comment, we would like to give more description of experimental details in ESI.

- **P9, para. 2:** The mint plants were born with cutting propagation technique and grown with water culture method; thus, we can't investigate the influence of Pb/Sn leakage to the early development of the mint plants. We analysed the toxicity of Pb/Sn leakage on the seed germination and seedling growth in the model species *brassica campestris* (cabbage). Our results show that the existence of heavy metal cations lead to the inability of seeds

germination in the first place and reduce seed growth (Figure S5). In the 35 and 250 ppm Pb/Sn contaminated soil, the seed germination and early seedling growth were reduced compared to control. Moreover, the seed grows much slowly in Sn cations contaminated soil, this could be because the Sn is known for its higher ability to acidify than Pb, since the toxicity of heavy metals to seed germination and seedling growth is known to be affected by soil pH.

- **ESI, P1, para 1 line 3:** The Pb/Sn exist in natural soil in several speciations: exchangeable form, carbonate bounded way, oxides form, organic form, and residual form.
- **ESI, P6-P7:**

3. Seed germination and early seedling growth

The mint plants were born with cutting propagation technique and grown with water culture method; thus, we can't investigate the influence of Pb/Sn leakage to the early development of the plants. We perform the experiment based on *Brassica campestris* (Cabbage) from seeds sown. And we found the existence of Pb/Sn perovskite and Pb_{12}/Sn_{12} lead to the inability of seeds to germinate in the first place and reduce the seed growth. In the 35 and 250 ppm Pb/Sn contaminated soil, the seed germination and seed growth were reduced compared to control. Moreover, the seed grows much slowly in Sn cations contaminated soil, this could be because the Sn is known for its higher ability to acidify than Pb, since the toxicity of heavy metals to seed germination and seedling growth is known to be affected by soil pH.

Figure S5. The photo of cabbage seed germination and seedling growth.

[a] Kjær, C., M. B. Pedersen and N. Elmegaard (1998). "Effects of Soil Copper on Black Bindweed (*Fallopia convolvulus*) in the Laboratory and the Field." *Archives of Environmental Contamination and Toxicology* 35(1): 14-19.

§ Is there any way to obtain a pure soil sample, (without ANY Pb in it). That would be the actual ideal reference point.

Response 2- 6: The plants can grow in lead-free water or quartz sand. In this case, the lead concentration will be below the detection limit. We are not sure how this system would be an ideal reference point since we will not be able to extract the change of lead uptake ability using Pb from perovskite and lead from other contaminants. See also the comment below.

§ The given equation for the uptake ability is only valid for the cases where there is contamination from perovskite, but not for the presence of Pb already in the soil from other sources than perovskite, which makes it quite unclear what can be learnt from it.

Response 2-7: The equation is valid for any contaminant. Indeed, it is commonly used in similar studies. If one would like to use the equation for the Pb already present in the natural soil it should grow the plants in lead-free soil to have a reference and then contaminate the lead-free soil. Unfortunately accessing a lead-free natural soil is rather complicated since the entire surface of the planet is lead contaminated by human activities. Removing lead from natural soil should be in principle possible but not common. What we are learning from the equation reported in the main text is that the uptake ability of lead from perovskite is xx% higher than the lead already present in natural soil.

§ on page 8, it is stated “...since Sn can rapidly oxidise to toxicologically inactive Sn compounds if dispersed in the environment” Where does the information about less toxic Sn⁴⁺ species come from? Is there a reference to support this? Some additional aspects that are unclear, incomplete, incorrect or inconsistent:

Response 2-8:

We agree with the reviewer that Sn⁴⁺ is not necessarily less toxic than Sn²⁺ or Pb²⁺. This is not what we mean. What we are trying to say is that if Sn²⁺ is dispersed in the environment, it will readily form oxygenated Sn precipitates, which have poor solubility in water. To make it more clearly, we have modified our wording as follows P8, line 3:

- **P8, para 1. line 2:** Among the others, tin-based perovskites are one of the most exciting possibilities²³, since if Sn²⁺ compounds are dispersed in the environment, they rapidly oxidise to more stable oxygenated Sn⁴⁺ compounds, which have poor solubility in water.

§ Given the nature of the experiment, it would be plausible to expect relatively large fluctuations in results. Unfortunately, it is not mentioned how many plants have been investigated for each condition. It is also not clear if the markers in Fig. 2 and 3 are individual data points, or statistical indicators of the box plots. If there would be only two or three plants per condition, it would cast a lot of uncertainty. I would strongly recommend adding all the data points in the plot, or at least mention the number of plants per condition explicitly.

Response 2-9:

In the updated manuscript ESI P4-5-part 2 mint plants growth, we have mentioned five plants were grown for each condition. We have added all the experiment data in ESI.

• ESI P10-11:

5. Experimental data:

Table S6: Pb uptake by mint plants grown in Pb perovskite contaminated soils

PbI ₂ +MAI	ROOT (mg/kg)		STEM (mg/kg)		LEAF (mg/kg)	
		average		average		average
250 mg/kg	4896.8	3905.83	178.99	209.55	426.79	405.73
	3416.6		240.11		384.67	
	3404.1		/		/	
35 mg/kg	119.51	130.39	6.73	10.53	/	/
	141.87		13.61		/	
	129.54		11.26		/	
5 mg/kg	28.77	26.85	4.13	4.69	8.82	9.97
	24.86		4.43		10.34	
	26.92		5.51		10.76	
Control (0 mg/kg)	9.32	11.99	3.88	3.36	8.44	7.97
	11.49		2.84		7.5	
	15.16		/		/	

Table S7: Pb uptake by mint plants grown in 5-ppm Pb perovskite and PbI₂ contaminated soils

	ROOT (mg/kg)		STEM (mg/kg)		LEAF (mg/kg)	
		average		average		average
Control (0 mg/kg)	9.32	11.99	3.88	3.36	8.44	7.97
	11.49		2.84		7.5	
	15.16		/		/	
PbI ₂ +MAI	28.77	26.85	4.13	4.69	8.82	9.97
	24.86		4.43		10.34	
	26.92		5.51		10.76	
PbI ₂	22.61	22.44	2.49	2.72	6.65	5.97
	19.84		2.66		6.14	
	24.9		3.03		5.13	

Table S8: Sn uptake by mint plants grown in Sn perovskite contaminated soils

SnI ₂ +MAI	ROOT (mg/kg)		STEM (mg/kg)		LEAF (mg/kg)	
		average		average		average
250	93.5	67.8	21.3	13.6	8.0	9.1

	37.5		10.2		18.7	
	76.4		14.7		6.0	
	71.3		13.5		3.8	
	60.5		8.2		132.0	
35	9.0	17.3	3.8	4.5	4.8	3.7
	30.8		8.1		3.7	
	13.4		2.5		4.4	
	25.5		4.3		2.9	
	7.8		1.8		2.6	
5	<1.6	6.5	3.4	3.6	<1.6	2.5
	3.3		4.5		1.6	
	10.1		2.9		3.5	
	2.9		<1.6		<1.6	
	9.6		2.4		<1.6	
Control	4.1	4.1	<1.6	2.4	<1.6	<1.6
	<1.6		<1.6		<1.6	
	<1.6		<1.6		<1.6	
	<1.6		2.1		<1.6	
	<1.6		2.8		<1.6	

Table S9: Sn uptake by mint plants grown in 35-ppm Sn perovskite and SnI₂ contaminated soils

	ROOT (mg/kg)		STEM (mg/kg)		LEAF (mg/kg)	
		average		average		average
SnI ₂ +MAI	9.0	17.3	3.8	4.5	4.8	3.7
	30.8		8.1		3.7	
	13.4		2.5		4.4	
	25.5		4.3		2.9	
	7.8		1.8		2.6	
SnI ₂	11.2	12.1	4.4	5.4	2.8	2.9
	11.6		5.3		3.1	
	17.9		7.7		3.5	
	9.9		7.5		<1.6	
	9.5		2.1		2.3	
Control	4.1	4.1	<1.6	2.4	<1.6	<1.6
	<1.6		<1.6		<1.6	
	<1.6		<1.6		<1.6	
	<1.6		2.1		<1.6	
	<1.6		2.8		<1.6	

§ on page 8 it reads: “Not surprisingly, we found that the organic cation has a substantial impact on the lead uptake ability of mint. The data reported in Fig. 2 show that the lead content in mint is similar or even lower than in plants grown in natural soil.” This last sentence is very unclear. What I think it meant is “The lead content in mint grown in soil contaminated with perovskite (PbI₂+MAI) is similar or even lower than in plants grown in natural soil.” If that is not what is meant, the authors should improve their wording and make things clear. If that is what is meant, it is incorrect, based on Figure 2, where the red “perovskite” data is always higher than the blue “PbI₂” data.

Response 2-10:

Thank you for noting this. We have modified our sentence as follows:

- **P7 line 3 from end:** The data reported in Fig.2 show that the lead content in mint grown in soil contaminated with the only PbI₂ is similar or even lower than in plants grown in natural soil.

§§ On the topic of this data, it is incomplete. Why is there no PbI₂ data for the 0 and 35mg/kg cases for Pb, and for the 0 and 5mg/kg cases for Sn?

Response 2-11:

0 mg/kg cases mean without any addition; thus, the mint grown in natural soil is the 0 mg/kg cases. During the experiment design, based on the literature about Pb/Sn uptake ability; for SnI₂, we thought the difference between control and 5mg/kg were not distinguished as that of control and 35 mg/kg, so we only designed 35mg/kg experiment.

§ The abstract mentions “standard lead contaminating soil”. I assume what is meant is “standard lead contaminatED soil”, but more importantly, this is not an appropriate term to use in the abstract of a manuscript dealing with such a sensitive topic. It suggests that soil is somehow contaminated by default, or that it is there naturally, which is not at all the case. Related to this, on page 3, it reads: “We measured the natural lead concentration of soil around 36 mg/kg,...”. The term “natural lead concentration” is deceptive. The lead concentration in the soil is non-zero because humans started to mine lead, with soil contamination as a result (reference should be made to: Babayigit et al. (10.1557/mre.2017.17)). Otherwise it would have never moved beyond the earth’s crust. Using the term “natural lead concentration” also creates the impression that, to some standard, a small increase in the concentration is unimportant, while it is in fact yet another undesirable increase on top of the unfortunate concentration mankind has already induced throughout history. The lead concentration could be termed as “effective”, but definitely not “natural”. All other instances of “natural” in the manuscript should also be removed for that matter. The context of the paragraph actually does not make it clear to me whether the authors are not aware of this or they simply have not expressed it explicitly. Whichever it may be, it creates a strong bias from the start, and it is important to refrain from the use of any phrase containing “natural lead” altogether, given the sensitivity of the subject at hand.

Response 2-12: Thanks for noting this, we have added the reference (Babayigit et al. (10.1557/mre.2017.17) in p3 para 2 line 1; and we have modified all the “natural lead” and “standard lead” in manuscript, as follows:

- **p3 para. 2 line 1:** Babayigit et al. <https://doi.org/10.1557/mre.2017.17>
- **P3 para. 2 line 2:** “natural lead concentration” ► “effective lead concentration of natural soil”
- **P6 Figure 2:** “natural lead-contaminated soil” ► “natural soil”
- **P6 para.1 line 8:** “natural lead concentration of the soil” ► “effective lead concentration of natural soil “
- **P9 para 3. Line5:** “natural lead-contaminated soil” ► “natural soil”

§ The authors fail to mention that for certain meat products the WHO maximum tolerable tin concentration is only 50mg/kg (http://www.fao.org/tempref/codex/Meetings/CCCF/CCCF5/cf05_INF.pdf)

Response 2-13: We thank you for noting this error. We have now corrected these values in P9 line 3/lin4.

- **P9 line1:** These values were far below the maximum tolerable levels proposed by the Food and Agriculture Organization of the United Nations²⁵, which were 150 mg/kg in canned beverages and 250 mg/kg in other canned foods (for certain meat products, the maximum tolerable tin concentration is 50mg/kg).

§ To the best of my knowledge, this is the first study that uses plants to evaluate the toxicity of such halide perovskites or degradation products thereof. On the other hand, the authors do make overstated claims, such as:

p2: “this work provides the first experimental evidence that we need to treat PSCs with exceptional care”.

p3: “This study provides the first set of experimental data to assess the environmental risk of halide perovskites”

p9: “we provided the first data set addressing the environmental impact of the lead-based halide perovskites”

Wherever and whenever this work could be published, these statements have to be removed or altered, as reference 9 and Benmessaoud et al. (10.1039/C5TX00303B) have also ventured in this direction, albeit with different approaches.

Response 2-14: we thank you for noting this. We have added the reference (Benmessaoud et al. (10.1039/C5TX00303B)) in P3 para.1 line 5 from the end. And to avoid the overstated claims, we have now:

- **P3 para. 1 line 8:** Benmessaoud et al. DOI: 10.1039/C5TX00303B
- **P2 para.2 line 5:** This work is the first time that use plants to evaluate the toxicity of perovskite or degradation products, and provides the experimental evidence that we need to treat lead PSCs with exceptional care”.
- **p3 end of para 1:** This study provides the first time of using plants to assess the environmental risk of halide perovskite, based on the experimental data of metal cations uptake by plants from the soil.
- **p9 para 3 line 1:** “we provided the data set that use plants to address the environmental impact of the halide perovskites”.

Finally, the authors conclude “We also proved that tin perovskites, which are often proposed as a safe alternative, are effectively safer than lead-based perovskite”. This statement lacks any form of nuance. I consider it highly inappropriate to draw this conclusion based on one set of experiments by one research unit, on one type of organism. At best the results give some indication that this could be the case, but a vast amount of additional work needs to be done to come to the overall conclusion as stated in the manuscript. Only one type of plant is used in one type of experiment, and while this is a necessary start, it is by no means an inclusive study. With this, and certainly considering the manuscript’s many flaws I addressed in this report, the conclusion of the manuscript is far too generalized and oversells the work. The study reports observations but fails to reveal any mechanism. It is interesting but for now raises more questions than it answers.

Response 2-15:

The metal uptake by plants is influenced by soil pH, the metal concentration in the soils, metal cation exchange capability between roots and soil, and other physic-chemical factors [a]. Mint plants are a rustic hyper-tolerant plant to several heavy metals (lead, tin, thallium, etc.) and have been used as a reference in heavy metal uptake study [b-d].

We agree with the reviewer that more data from different plants can help to strengthen the conclusion of our study. Thus we performed additional experiments based on capsicum annum L (chilli, which is a low Pb accumulating capability plant) and brassica campestris L. ssp (cabbage, which is a middle Pb accumulating capability plant). Most of the chili and cabbage plants died in soil with 35 and 250 ppm Pb contamination (as shown in Figure S6). For the survived plants, both the chilli and cabbage show the same trend obtained from the mint plants, i.e. the uptake ability of lead from perovskite is higher than the lead already present in natural soil.

We have edited our manuscript as follow:

p9 para 3 line 5: We also proved that the tin perovskites, which are often proposed as a safe alternative, are more resilient than lead perovskites to enter into plants and thus into the food chain.

ESI Part 4. Capsicum annum and Brassica campestris as alternatives to mint plants

The metal uptake by plants is influenced by soil pH, the metal concentration in the soils, metal cation exchange capability between roots and soil, and other physic-chemical factors [a]. Mint plants are a rustic hyper-tolerant plant to several heavy metals (lead, tin, thallium, etc.) and have been used as a tester in heavy metal uptake study [b-d]. In this work, we investigated the plant absorption of heavy metals from the perovskite-contaminated soil. We agree the mint cannot completely represent all plants uptake of the metal element, but this protocol still creates a reasonable, reliable and repeatable standard that enables to obtain test results efficiently.

Besides the mint plants experiment, we performed additional experiment based on capsicum annum L (chilli, which is a low Pb accumulating capability plant) and brassica campestris L. ssp (cabbage, which is a middle Pb accumulating capability plant) as examples. Both the chili and cabbage died at 35 and 250 ppm (as shown in Figure S6). For the survived plants, both the chili and cabbage show the same trends as the results obtained from the mint plants.

For the chilli: After adding 5 mg/kg Pb²⁺ perovskite in soil, the mean lead concentration in mint roots and stems showed a significant increase up to 9.1 mg/kg. Considering that the lead concentration in natural soil was 36.3 mg/kg, which means we only add extra 13.8% Pb²⁺ in native soil, but the lead concentration increased about 60% compared to that grown in natural soil.

For the cabbage: After adding 5 mg/kg Pb²⁺ perovskite in soil, the mean lead concentration in cabbage plants showed a significant increase up to 11.2 mg/kg. Considering the lead concentration in natural soil is 36.3 mg/kg, which means we only add extra 13.8% Pb²⁺ in natural soil, but the lead concentration increased up to 98.9% than in plants grown in native soil.

Figure S6. Photo of chili experiment.

Figure S7. Photo of cabbage experiment.

Figure S7. Lead concentration in different parts from mint plants grown in 5 ppm perovskite (PbI₂+MAI)/PbI₂ contaminated soils and natural soil. Note: to get enough weight for Pb concentration measurement, for the chilli, only roots and stems are used for characterisation; and the whole cabbage plants were used for the Pb concentration characterisation.

Table S10. Pb concentration in Chili.

Pb ²⁺ added (mg/kg)	Roots +Stems (mg/kg)	
	PbI ₂	PbI ₂ +MAI
5	7.48	11.7
	9.62	8.35
	8.44	7.86
	7.49	9.70
	N/A	7.89
Control	6.79	

	5.71
	4.85
	5.37

Table S11.Pb concentration in cabbage.

Pb²⁺ added (mg/kg)	Pb in plant (mg/kg)	
		9.11
5	9.49	8.92
	11.2	10.4
	12.1	12.0
	8.60	9.63
	6.83	
Control	5.26	
	5.37	
	5.05	
	9.11	

Ref.

[a] Zeng, F., et al., The influence of pH and organic matter content in paddy soil on heavy metal availability and their uptake by rice plants. *Environmental pollution*, 2011. 159(1): p. 84-91.

[b] Zheljzkov, V.D., L.E. Craker, and B. Xing, Effects of Cd, Pb, and Cu on growth and essential oil contents in dill, peppermint, and basil. *Environmental and Experimental Botany*, 2006. 58(1-3): p. 9-16.

[c] Ghasemidehkordi, B., et al., Tin Levels in Perennial and Annual Green Leafy Vegetables. *International Journal of Vegetable Science*, 2017. 23(4): p. 340-345.

[d] Prasad, A., et al., Effect of chromium and lead on yield, chemical composition of essential oil, and accumulation of heavy metals of mint species. *Communications in soil science and plant analysis*, 2010. 41(18): p. 2170-2186.

REVIEWERS' COMMENTS:

Reviewer #1 (Remarks to the Author):

I am satisfied with all the corrections the authors made regarding my concerns and am pleased to recommend acceptance for publication.

Gary Hodes

Reviewer #2 (Remarks to the Author):

The authors have adequately resolved my concerns. The manuscript has improved substantially. I have only one remaining remark:

The main text has been altered on p9 paragraph 3 line 5: "We also proved that the tin perovskites, which are often proposed as a safe alternative, are more resilient than lead perovskites to enter into plants and thus into the food chain."

The word "resilient" is a poor choice. It is confusing and could even suggest the opposite of what is meant. It would be better to say that tin perovskites "have a lower tendency" to enter into plants, or are "less likely" to do so.

With this final tweak, the manuscript can be published.